# Evaluation of Multivariate Filters on Vibrational Spectroscopic Fingerprints for the PLS-DA and SIMCA Classification of Argan Oils from Four Moroccan Regions

**DOI:** 10.3390/molecules28155698

**Published:** 2023-07-27

**Authors:** Meryeme El Maouardi, Mohammed Alaoui Mansouri, Kris De Braekeleer, Abdelaziz Bouklouze, Yvan Vander Heyden

**Affiliations:** 1Biopharmaceutical and Toxicological Analysis Research Team, Laboratory of Pharmacology and Toxicology, Faculty of Medicine and Pharmacy, University Mohammed V, Rabat 10100, Morocco; meryeme.el.maouardi@vub.be (M.E.M.);; 2Department of Analytical Chemistry, Applied Chemometrics and Molecular Modelling, Vrije Universiteit Brussel (VUB), Laarbeeklaan 103, 1090 Brussels, Belgium; 3Nano and Molecular Systems Research Unit, University of Oulu, FIN-90014 Oulu, Finland; 4Pharmacognosy, Bioanalysis & Drug Discovery Unit, Faculty of Pharmacy, University Libre Brussels, 1050 Brussels, Belgium

**Keywords:** Argan oil, mid-infrared spectroscopy, near-infrared spectroscopy, chemometric tools, classification models, multivariate filters

## Abstract

This study aimed to develop an analytical method to determine the geographical origin of Moroccan Argan oil through near-infrared (NIR) or mid-infrared (MIR) spectroscopic fingerprints. However, the classification may be problematic due to the spectral similarity of the components in the samples. Therefore, unsupervised and supervised classification methods—including principal component analysis (PCA), Partial Least Squares-Discriminant Analysis (PLS-DA) and Soft Independent Modeling of Class Analogy (SIMCA)—were evaluated to distinguish between Argan oils from four regions. The spectra of 93 samples were acquired and preprocessed using both standard preprocessing methods and multivariate filters, such as External Parameter Orthogonalization, Generalized Least Squares Weighting and Orthogonal Signal Correction, to improve the models. Their accuracy, precision, sensitivity, and selectivity were used to evaluate the performance of the models. SIMCA and PLS-DA models generated after standard preprocessing failed to correctly classify all samples. However, successful models were produced after using multivariate filters. The NIR and MIR classification models show an equivalent accuracy. The PLS-DA models outperformed the SIMCA with 100% accuracy, specificity, sensitivity and precision. In conclusion, the studied multivariate filters are applicable on the spectroscopic fingerprints to geographically identify the Argan oils in routine monitoring, significantly reducing analysis costs and time.

## 1. Introduction

The Argan tree (*Argania spinosa* L. Skeels) is an endemic plant that represents the only species of the genus *Argania* and the family *Sapotaceae* in Morocco [1]. The Argan forest, which currently covers an area of 800,000 ha [2], was recognized in 1998 as a UNESCO international biosphere reserve [3]. It has a significant impact on economic and social development in Morocco, and it is frequently the only source of income for certain local communities. The Argan kernels provide a precious oil that is rich in fatty acids, tocopherols, sterols and anti-oxidant compounds (especially vitamin E) [1]. In addition to its high nutritional value, several studies demonstrated its potential role in many activities, such as in blood pressure reduction [4], anti-inflammatory [5], antiproliferative [6], cardioprotective [7], antidiabetic [8], antioxidant [9] and hypolipidemic [10] activities, resulting in the prevention of several diseases. The Argan oil’s composition determines its intrinsic quality and could be influenced by several factors, such as cultivar and environment, which affect the fruit physiology [11]. Other factors, such as latitude, climatic conditions [12], fruit maturity [13], harvesting [14] and extraction technology [15], could also affect the qualitative and quantitative composition of Argan oil.

Argan oil has seen an increase in national and worldwide demand in recent decades, as well as an increase in its falsifications. As a result, its authentication has become a crucial responsibility to maintain its quality, safety, and legitimacy, as well as to guarantee the safety of customers. Physicochemical properties, such as the acidity index of saturated and unsaturated fatty acids, triglyceride-, triacylglycerol-, sterol-, and phenolic compound contents, have been used to investigate the effect of geographic origin on the qualitative profile of Argan oil [16]. Traditionally, these parameters are estimated by classical analytical methods, which are frequently based on chromatographic techniques (gas chromatography and high-performance liquid chromatography) [17]. Nevertheless, these methods are of low speed, high cost and they need sample preparation. In the last few years, attention on authentication and quality control has been focused on using spectroscopic techniques, such as mid-infrared (MIR) [18], near-infrared (NIR) [19] and Raman spectroscopy [20], that offer fast, non-destructive, and cost-effective methods for food analysis. These spectroscopic techniques have been widely used in research studies related to food control. They have become authoritative analytical tools in the study of edible oils and fats. Various studies have been published using spectroscopic techniques to differentiate edible oils according to their origin [18,21,22,23]. Moreover, spectroscopic techniques were also used for the detection of adulteration [24,25]. All of these studies applied spectroscopic techniques as useful tools to assess the origin or the quality, and to determine the adulteration of edible oils and fats. However, these analytical techniques generate more complex data, which need chemometric data-handling tools to extract useful information [26]. Various chemometric techniques, such as principal component analysis (PCA), Partial Least Squares-Discriminant Analysis (PLS-DA), Linear Discriminant Analysis (LDA), Partial Least Squares Regression (PLS), and Soft Independent Modeling of Class Analogy (SIMCA), are commonly used to extract useful information from the data matrix (**X**) and to link it to properties of the oils.

Although these multivariate data analysis methods compress the instrumental variables into a few informative components (latent variables), spectroscopic data still exhibit many correlated variables due to wide bands and overtones. In order to attenuate the covariance of spectra, multivariate filters may be applied before model building [27]. The objective of these multivariate filters is to remove or attenuate the unwanted covariance structure of matrix **X**, that is not correlated with vector **y** [27]. As a result of the removal of the unwanted variability, the complexity of the model is reduced and the resulting data contain the remaining covariance patterns which passed through the filter and are, ideally, useful for modeling. In addition, as these filters increase the sensitivity and specificity of the models, less latent variables are generally required to obtain a high percentage of explained variance. Different studies have shown improvements in the predictive capacity of classification or regression models using the multivariate filtering methods: Orthogonal Signal Correction (OSC), Generalized Least Squares Weighting (GLSW) and External Parameter Orthogonalization (EPO) [27,28,29,30,31].

In this perspective, this work aimed to evaluate the application of standard preprocessing methods, such as autoscaling, Multiplicative Signal Correction (MSC) and derivatives, as well as of multivariate filters (OSC, GLSW and EPO) on two vibrational spectroscopic data sets, NIR and MIR data, for the discrimination of Argan oil samples from four geographical regions. In this evaluation, the PCA model, as a visualization and unsupervised classification method, was applied, while PLS-DA and SIMCA, as supervised pattern recognition techniques, were compared.

## 2. Results and Discussion

### 2.1. NIR and MIR Spectral Characteristics of Argan Oil

Before chemometric analysis of the NIR and MIR spectra, the spectral characteristics of Argan oil samples were interpreted to identify the different functional groups in the samples. The wavenumbers observed in the spectra and the related functional groups are summarized in Table 1.

The NIR fingerprint of Argan oil (Figure 1a) is characterized by some bands: 8300–8200 cm^−1^ is due to second overtones of C–H stretching vibrations; 7180 to 7075 cm^−1^ to the C–H combination band; 6000–5500 cm^−1^ to the first overtone of C–H stretching vibrations from –CH_2_ and –CH_3_ functional groups, and 4800–4500 cm^−1^ may be related to the first overtones of C–H stretching from –HC=CH– groups of unsaturated fatty acids (linoleic acid C18:2). Small peaks associated with the combination bands of C–H and C–O stretching vibration appear between 4350 and 4210 cm^−1^. This observation is consistent with other study findings [32].

The MIR spectra recorded present two important regions (3010 to 2700 cm^−1^ and 1750 to 600 cm^−1^) where some dominating peaks may be identified (Figure 1b). The band around 3008 cm^−1^ is for stretches of –C=C–H (cis) groups. Bands arising from asymmetric and symmetric CH_2_ stretching are seen at 2923 and 2854 cm^−1^; C=O stretching vibrations are at 1744 cm^−1^; CH_2_ and CH_3_ scissoring vibrations 1463 and 1377 cm^−1^; C–O stretching vibrations occur at 1238, 1160, and 1118 cm^−1^; and the –HC=CH- (cis) rocking mode corresponds to a 722 cm^−1^ signal, which characterizes a long chain of unsaturated fatty acids [32].

### 2.2. PCA Analysis

As shown in Figure 1, the NIR and MIR spectra of the 93 Argan oil samples are very similar because of their similar chemical composition. No clear differences between the four geographical origins of the Argan oils were observed. In order to explore the similarities and differences among and between the Argan oil groups, PCA analysis was carried out. The MIR and NIR spectra were pretreated using different preprocessing methods, such as SNV, autoscaling, mean centering and derivatives. However, because of the strong similarity between the samples, no clear distinction between the four regions was observed after applying the standard preprocessing methods. Therefore, the multivariate filters were applied on the NIR and MIR data to remove the interfering within-groups contributions from the data and to increase the variation between the classes [27].

Figure 2a shows the MIR PCA score plots for the Argan oil samples from different regions after applying an OSC (with 5 components), EPO (with 18 PCs) or GLSW (with an alpha parameter of 0.0002) multivariate filter. The MIR results were optimized by selecting the spectral ranges 3238–2696 cm^−1^ and 1863–400 cm^−1^, which contain characteristic information of the Argan oil. The OSC PCA score plot explained 87% of the total variance by the first two PCs and provided separation tendency between the four regions. The EPO and GLSW PCA score plots (both explaining only 28% of the total variance by the first two PCs) show similar results with four well-separated clusters, and each one representing an Argan oil region. The low explained variance (~28%) by the first two principal components is due to the application of these multivariate filters which focus on the variability between classes as explained in the Material and Methods Section. The same tendency was observed in a previous application of these preprocessing methodologies [33]. In the plot, PC1, which explained 16% of the total variance in the data, separates the continental samples (provinces Tiznit and Taroudant) from the coastal samples (Essaouira and Agadir). Analyzing the PC plots, some samples were suspected to be outliers to the other class members. Therefore, Hotelling’s T2 vs. Q residuals plots, with a threshold of 95%, were used to check the homogeneity of the dataset and to detect whether any outliers occur. According to the obtained Hotelling’s T2 vs. Q residuals plots, these samples fall inside the threshold. Therefore, they were not considered as outliers.

PCA was also applied on the full NIR spectral data. Figure 2b displays the PC1-PC2 score plots of the four regions of Argan oil after applying a filter method. The OSC PCA score plot explained 57% of the total variance; however, no distinction between the regions was observed. Figure 2b also shows the EPO and GLSW PCA score plots which account for about 53 and 60% of the total variance, respectively. The Tiznit and Taroudant classes are clearly distinguished from the Agadir and Essaouira samples, mainly along PC2, which accounts for 22 and 17% of the variation in the EPO- and GLSW-treated data, respectively. In addition, Tiznit and Taroudant, the two continental provinces, were situated closer to each other than to the other regions.

In order to evaluate the performance of these three algorithms (OSC, EPO and GLSW), a new data set was created, where the spectra were randomly assigned to classes. Then, PCA models were created for both MIR and NIR data sets. This step is an evaluation to make sure that these algorithms allow distinguishing the groups based on the real variation between the spectra and not on chance correlations. The PC1–PC2 score plots are displayed in the Appendix A. For none of the three multivariate filters, a distinction between the four groups was observed, as was expected and hoped. These results, from both instruments, show that OSC, EPO and GLSW could be applied in practice and do not show groups due to chance correlations.

The outcomes from the NIR and MIR PCA plots, obtained after EPO and GLSW preprocessing, are consistent with previous research conducted by Kharbach et al. [34] and Elgadi et al. [35], who classified Argan oil from five geographical origins. In these studies, the chemical composition of Argan oil was analyzed using PCA and PLS-DA. Their findings also revealed that samples from the Tiznit and Taroudant regions are somewhat closer, which was explained by the similarity of the geographical parameters. Taous et al. [36] used a non-destructive sampling approach to distinguish the geographic origin of Argan oil with MIR spectroscopy and chemometrics. Similar outcomes were found, confirming the link between the geographic origin and group distribution in the PCA plots. According to Appendix A Taroudant and Tiznit, two continental cities, have the lowest rainfall (210 and 205 mm/years respectively) and the highest altitude, and both are further from the sea than Essaouira and Agadir. In addition, these continental locations were characterized by high concentrations of α-tocopherol, δ-tocopherol, and β-tocopherols, which may explain the distribution of the clusters in the PC plots. This also confirms that several factors, including the geographical origin of fruits, altitude, climate and environmental conditions, can affect the chemical composition of the resulting Argan oil.

In summary, based on the distribution of the clusters in the PC plots, the PCA results showed that EPO and GLSW preprocessing provides better results than OSC. However, both MIR and NIR spectroscopic techniques provide comparable outcomes to those obtained with the chemical analysis of Argan oil.

### 2.3. PLS-DA and SIMCA Classification Models after Different Preprocessings

#### 2.3.1. PLS-DA Models on MIR Data

To discriminate between the four regions of Argan oil, PLS-DA models were built for both the MIR and NIR spectra. PLS-DA models were first developed on the raw spectra. Then, different preprocessing techniques (see above) were applied to find the best model with good classification results. The number of significant latent variables and the predictive abilities of the PLS-DA models developed were evaluated by the Venetian-blinds cross-validation. The quality parameters, precision, specificity, sensitivity and accuracy, were assessed.

Table 2 provides a summary of the performance parameters for the MIR PLS-DA models. Considering the MIR spectra, without preprocessing, the best PLS-DA model, built with 7 LVs, was unable to discriminate the different regions. Several test set samples could not be assigned to their proper group. Thus, they were associated with an accuracy between 77 and 85%, a range of specificity between 81 and 97%, a sensitivity from 50 to 82%, and a precision between 45 and 85%.

Different PLS_Toolbox built-in preprocessing techniques, such as autoscaling, mean centering, SNV, multiplicative signal correction and derivatives, have been tested. Most of the resulting models were best, with an accuracy higher than 0.82 when containing between 10 and 20 LVs. However, a number of LVs in this range may be risky because unwanted information or noise can be included in the model and cause overfitting. The best combinations of preprocessing that gave the highest accuracy values with the smallest reasonable LVs number are listed in Table 2. It was found that autoscaling improved the PLS-DA model with a maximum value of accuracy of up to 95% for the test set. However, a more significant improvement was observed after the application of the 2nd derivative (order 2, window 15 pt) followed by mean centering. Figure 3a shows the classification error used to choose the number of LVs in the model. This parameter provided information on the average calibration and cross-validation errors, which decreased with model complexity. It turned out that the classification can be achieved for the PLS-DA model with at least nine LVs. We want to keep the model as simple as possible and do not want to apply more LV, as this could lead the model to become progressively overfitting. The classification accuracy increased up to 100% using the 2nd derivative followed by mean centering as preprocessing. The specificity and sensitivity values ranged between 95 and 100% and the precision between 94 and 100%, which means that some samples were not well classified for some regions. The prediction models in Figure 3 illustrates these findings. As represented, the classification threshold in the established PLS-DA model for differentiating one class from the other classes was determined as 0.5 (red dotted line). Samples that show a calculated response higher than 0.5 are classified as belonging to the target class, while samples that show lower values are rejected. The Taroudant and Tiznit groups (Figure 3d,e) were well classified, which explains the 100% specificity, sensitivity and precision. However, quite a large internal class dispersion was observed (predicted values from 0.5 to 1.4). Meanwhile, there are some misclassified samples in the Essaouira and Agadir clusters (Figure 3b,c), which explains the lower precision, specificity and sensitivity values obtained.

From these results, we conclude that there is a large similarity between samples from the four regions, which did not allow to correctly discriminate the four groups using the common preprocessing methods. Trying to improve the model discrimination abilities for all classes, multivariate filters were applied. Three multivariate filters (OSC, EPO and GLSW) were applied to check whether they successfully reduce the internal class dispersion and, at the same time, improve the model predictivity and reliability.

The OSC filter, with 5 components, provided its best result in a model with five LVs (Figure 4a), which noticeably improved the classification accuracy to 100% for both training and test sets (Table 2). The constructed OSC-PLS-DA model allowed classifying correctly all samples. In this model, the specificity, sensitivity and precision were 100% for all classes and the classification threshold was determined at 0.5 with quite a small internal class dispersion (Figure 4b). Similar PLS-DA prediction results were obtained for the four models, which is why only the Essaouira prediction results obtained with the three multivariate filters were displayed in Figure 4b.

Moreover, the constructed GLSW-PLS-DA and EPO-PLS-DA models also reached very good results with excellent predictive abilities in classifying the samples from the four regions. The optimal number of LVs was three for both EPO and GLSW multivariate filters, (Figure 4a). As presented in Table 2, the test set samples were clearly assigned to their proper classes with a specificity, sensitivity, precision and accuracy of 100%. As it is clear from Figure 4b, the target class, Essaouira, is predicted around 1 and the other classes are around 0. In addition to the classification threshold that does not exceed 0.4, the internal dispersion of the classes reduced (predicted values between 0.9 and 1.1). As a conclusion, the multivariate filter methods positively influenced the PLS-DA models on the MIR data.

#### 2.3.2. PLS-DA Models on NIR Data

PLS-DA analysis was also performed on the pretreated NIR data to select the best preprocessing method. Table 3 summarizes the classification performances with and without preprocessing. As expected, the unprocessed data provided an unsatisfactory classification on the test data with 54–97% sensitivity, 86–97% specificity, 35–94% precision and 81–97% accuracy.

Therefore, the challenge again was to find a suitable preprocessing approach to achieve the most accurate classification. After trying the often-applied preprocessing methods, including MSC, SNV, normalizing, Savitsky–Golay smoothing and derivatives, most of them resulted in models with an accuracy between 40 and 61%, and a number of LVs between 10 and 23. However, when constructing the PLS-DA model after the application of the MSC followed by the 2nd derivative (order 2, window 5 pt), the PLS-DA plot for each class resulted in a rather good classification. The optimal number of LVs retained based on the classification error was six (Figure 5a). The test set samples of the Taroudant, Essaouira and Tiznit groups were correctly attributed to their groups, which provides 100% specificity and sensitivity for these groups. However, one target sample from the Agadir test set was assigned as non-target sample, which resulted in a sensitivity of 96% for the Agadir test set (Table 3). Moreover, for the four classes, a significant internal dispersion in the classes was observed (with predicted values from 0.4 to 1.5). It could mean that the centroid of the classes was not very clustered, and the samples were spread far from their center [37].

OSC, GLSW and EPO multivariate filters were also applied to the NIR data before building the PLS-DA models. The selected numbers of LVs retained were eight for the model after OSC filtering and three for both EPO and GLW filters (Figure 6a). The classification results in terms of precision, sensitivity, specificity and accuracy for the training and test sets of the four groups are summarized in Table 3. The constructed models allowed correctly classifying all samples. The model sensitivity, specificity, precision and accuracy were 100% for all classes. Analyzing the sample predictions, a good class separation was observed for the preprocessing methods, mainly for EPO and GLSW (Figure 6b). These PLS-DA models show in Figure 6b that the training set from the Essaouira class was separated from the other classes, and the results of the test set confirmed the excellent discrimination. The model values predicted for the Essaouira class samples obtained with the tree multivariate filters were around 1 (which means that they were attributed to the class Essaouira), while the other samples were close to 0 (which means that they were indicated not to belong to that class). Therefore, a very certain and reliable classification of the Essaouira class samples is seen. Also the samples from the classes Agadir, Tiznit, and Taroudant were clearly separated from the other classes in the equivalent Figures.

To summarize, considering the number of LVs and the intragroup dispersion, the GLSW- and EPO-based models showed the best results, which, considering also the MIR results, showed that the performance of the three methods may be ranked as GLSW = EPO > OSC.

These results make it evident that the PLS-DA models obtained for the NIR and MIR datasets using multivariate filters as preprocessing techniques allow predicting training and test samples of the discriminated class from the three other classes. These results also demonstrate that the NIR and MIR fingerprints contain chemical information that allows PLS-DA to distinguish between the four classes of Argan oil. PLS-DA, in combination with NIR or MIR spectroscopic fingerprints and proper data pretreatment, may thus be applied in routine monitoring of Argan oil quality control.

It is known that Orthogonal Signal Correction methods or multivariate filters, in general, can lead to obvious overfitting when applied to the spectra forming the training set [38]. For this reason, the reliability of the three multivariate filter methods was evaluated by testing their performance on the data set with the randomly assigned classes (see PCA section above), to check for the occurrence of good classification models based on chance correlations. The NIR and MIR PLS-DA models from the latter data set obtained were not able to correctly classify samples to their predefined groups (Appendix A). Hence, the performance of the three multivariate filters seems appropriate.

#### 2.3.3. SIMCA Models

Another geographic discrimination approach was performed on the NIR and MIR datasets using SIMCA analysis to build models that should be capable of classifying each sample into its own group. The spectra were treated by different preprocessing methods (SNV, MSC, mean centering, 1st and 2nd derivative) in order to determine the most suitable. Table 4a shows only the best results obtained, after applying MSC + 1st derivative, EPO and GLSW preprocessing on the MIR data. The MSC + 1st derivative (order 2, window 15 pt) provided satisfactory results for the training set with a specificity of 100%, sensitivity between 77–92%, precision 97–100% and accuracy varying between 87–94%. The test set parameters, however, were insufficient, producing an accuracy between 76 and 84% and a sensitivity between 45 and 70%, indicating that the samples were not evenly distributed across their respective specified groups.

The GLSW (α = 0.0002) and EPO (PCs = 18) multivariate filters provided 100% specificity, sensitivity and accuracy in the classification of Argan oils for the training set, indicating that samples were correctly distributed in their proper groups. In the test set, the specificity was 100%, indicating that the samples which did not belong to the relevant region were correctly rejected by the models. However, the sensitivity was between 60–78% and the accuracy was between 80–99% for the test set, meaning that some samples were not correctly attributed to their respective groups.

Similarly, Table 4b exhibits the classification results for different pretreatments applied to the NIR data using the full spectral range. The GLSW and EPO filters showed a very satisfactory prediction performance with 100% precision, sensitivity, specificity and accuracy for the training set. However, for the test set, the developed SIMCA models produced a sensitivity between 81–92%, specificity of 100%, precision between 87–95% and an accuracy between 95–98%. According to the confusion matrix, 13 samples from the GLSW model and 11 samples from the EPO model were not assigned to their class, reducing the sensitivity and accuracy.

## 3. Material and Methods

### 3.1. Sample Collection

In 2021, 93 Argan oil samples were collected from four Moroccan regions, Taroudant (TA), Essaouira (ES), Agadir (AG) and Tiznit (TZ) (Appendix A). These regions are located in the arid and semi-arid areas of the southwest [39]. Temperature and rainfall data were collected from different weather stations for the period from 1989 to 2021 (Appendix A). All Argan oil samples were extracted mechanically from roasted kernels, produced by the women’s cooperative markets, and stored in the dark until the spectroscopic analysis. For each sample, three independent samplings were prepared and scanned in three different days. Two were placed in the training set and one in the test set. A total of 279 spectra were collected. Since the number of samples per class is not that large, the spectra from the first two days were kept for the training set and the last for the test set. These test set results, as also for the cross-validation results, may give a somewhat too-positive prediction compared to a completely independent test set. However, our approach still allows us to compare different preprocessing and/or classification techniques.

### 3.2. Spectroscopic Techniques and Spectrum Acquisition

Near-infrared (NIR) spectral data were collected in transflexion mode using the NIRA accessory of an FT-IR/NIR spectrometerᵀᴹ (PerkinElmerᵀᴹ, Waltham, MA, USA) and applying small glass petri dishes as a sample holder with a reflector placed in the axis of the light beam. The spectral data were collected over the range 10,000–4000 cm^−1^ (resolution 16 cm^−1^, accumulations: 32 scans).

A universal ATR accessory (UATR) was applied on the FT-IR/NIR spectrometerᵀᴹ (PerkinElmerᵀᴹ) to collect mid-infrared (MIR) spectra in the diffuse reflection mode within the range 4000–400 cm^−1^ at room temperature (resolution 4 cm^−1^, accumulations: 16 scans). The oil samples were positioned on a horizontal diamond crystal UATR plate. The MIR spectra were converted into vectors of 1613 variables and the combination of the vectors resulted in the **X** matrix with dimensions 279 by 1613.

NIR and MIR spectral acquisition were performed using PerkinElmer Spectrum software (version: 10.5.3.738).

### 3.3. Chemometric Tools for Classification

Chemometric tools extract information from multivariate data using mathematical and statistical methods. Some chemometric methods, such as Linear Discriminant Analysis, Support Vector Machine, Partial Least Squares-Discriminant Analysis and Soft Independent Modeling of Class Analogy, create mathematical models determined by the degree of similarity between two or more samples, to distinguish between two or more groups of observations (samples) and to determine to which group a given sample belongs to.

#### 3.3.1. Preprocessing Methods

Different data preprocessing techniques, such as autoscaling, Multiplicative Scatter Correction (MSC), Standard Normal Variate (SNV) and derivatives, can be used in order to remove sources of variation that do not carry relevant information. Multivariate filters, such as Generalized Least Squares Weighting (GLSW), External Parameter Orthogonalization (EPO) and Orthogonal Signal Correction (OSC) are less used for preprocessing, but they are extremely useful for eliminating baseline shifting and increasing signal-to-noise ratios [40]. The following preprocessing methods were evaluated in our modeling: mean centering, MSC, Savitzky–Golay smoothing plus derivatives, GLSW, EPO and OSC.

Autoscaling aims to give all variables an equal opportunity of being modeled by subtracting, column-wise, the mean value from each variable and then dividing the result by its standard deviation [41]. Each variable gets the same weight in the subsequent modeling.

MSC is one of the most used normalization transformations. Its goal is to make all samples appear to have the same scatter level at all wavelengths by removing the influence of the scatter from the spectra. To estimate and correct the scatter of the spectra, a reference spectrum, usually the average spectrum of a representative data set, is used [42].

Derivatives (first and second) of the spectra have been used in analytical spectroscopy for decades for their capability to remove additive effects and enhance small differences between similar spectra. The first derivative removes the additive baseline, and the second derivative removes the linear baseline. To avoid reducing the signal-to-noise ratio and to remove the influence of noise in the corrected spectra, both derivation methods first apply smoothing, which is done by the Savitzky–Golay algorithm [43].

The mean centering preprocessing approach is one of the most used preprocessing methods. It removes a systematic difference by subtracting the mean value of each variable, while maintaining similarities in the data set. After this processing, each row of the mean-centered data matrix shows how it differs from the data matrix’s average sample [30,31].

The concept of Orthogonal Signal Correction was introduced by Wold et al. [44] The goal of this spectral preprocessing method is to remove one or more directions in **X**, orthogonal to the **Y** and that account for the largest variation in **X**. OSC is performed as a preprocessing step to improve the calibration model.

External Parameter Orthogonalization aims to eliminate, from the **X** space, the part mostly affected by the external parameter variations such as the temperature or moisture effect. This algorithm, which is based on principal component analysis, finds the regions of the spectra that are affected by the external parameters and projects the spectra orthogonal to this variation; in this way the unwanted variations are removed [29].

Although GLSW was first designed for calibration transfer, it can also be used prior to classification and regression modeling. This technique aims to calculate a filter matrix to down-weight the external disturbances, i.e., temperature. It is also used to down-weight the differences between two instruments (in calibration transfer) to make them appear more similar and to achieve the best calibration and prediction errors with less latent variables in the subsequent modeling. The GLSW uses samples with similar **Y**-block values to identify the sources of variance and then filter it out from the original data [29,30].

Every preprocessing strategy aims to reduce the un-modeled variability in the data. Thus, the aim is to employ the appropriate preprocessing methodology, prior to model building and validation. It may be challenging to determine which preprocessing method is the best. However, combining several preprocessing methods is possible, but not advised, and the preprocessing techniques are expected to reduce the effective model complexity compared to the raw data modeling [45].

#### 3.3.2. Unsupervised Pattern Recognition

Principal component analysis is an unsupervised technique that transforms the original data space into a space of latent variables (principal components) which are orthogonal and each explains the maximal remaining variance in the dataset. The ***X*** matrix, representing the spectra of the different samples, is decomposed into two matrices, ***T*** of scores and $P^{T}$ of loadings (Equation (1)) [41,46]
(1)$$X=TP^{T}+E_{X}$$
where $E_{X}$ is the residual matrix. PCA was performed on each data set (MIR and NIR spectral data) to explore the association between groups and to detect occasional outliers.

#### 3.3.3. Supervised Pattern Reconstruction

PLS-DA is a well-known multivariate discrimination technique that depends on the ***X*** matrix (MIR or NIR spectra of Argan oil) and the **y** vectors (classes of geographical origin) to develop discriminative models. This supervised technique reduces the data to scores and loading matrices which allow one to find the most optimal number of latent variables. In PLS-DA, **y** contains qualitative information that identifies different classes of Argan oil. Two codes are created, “1” attributed to the target class that must be discriminated from the other alternative classes that all get the value “0”. Each class is once considered the target class; therefore, four models will be built [47].

Soft Independent Modeling of Class Analogy is a class-modeling technique widely used in chemometrics, which considers different classes individually modeled by a separate principal component analysis. The number of significant PCs is determined for each class of the training set by a Venetian-blinds cross-validation. The class distance can be calculated as the geometric distance from the principal component models. For visualization purposes, the SIMCA results can be graphically represented as a plot of the loadings and of the scores of the PCA, thus providing information about outliers, sub-groupings, and within-class structure [48].

Prior to the development of the PLS-DA and SIMCA models, the data were divided into a training set containing 186 of the spectra, measured on a first and second day, while the remaining 93 spectra were used as a test set on a third day. Several statistical parameters were used, for each class separately, to evaluate the fit and predictive performance of the PLS-DA and SIMCA models. The following statistical parameters are used:

Sensitivity, or the true positives (TP), defined as the percentage of the samples correctly attributed to the target class, which expresses the ability of the model to correctly recognize samples belonging to the considered class [49].

Specificity, or the true negatives (TN), known as the percentage of the samples that do not belong to the target class and which are indeed classified as the alternative class [49]. It describes the model capability to correctly reject samples belonging to all the other classes.

Precision is the number of correctly classified positive samples divided by the number of samples predicted as positive.

Accuracy, or the percentage of correct classification, represents the ability to correctly distinguish between the different samples. It is calculated based on the proportion of true positives and true negatives in all evaluated samples [49].

All of these parameters can take values between 0 (0%) and 1 (100%) and may be calculated referring to the training and test sets. The higher the accuracy, precision, specificity and sensitivity values are, the better the robustness of the model is [50].

#### 3.3.4. Software

Chemometric data analysis and processing were performed using MATLAB software 18.b (The Math-Works, Natick MA, USA) and PLS_Toolbox software version^®^ 8.6.1 8.2.1 (Eigenvector research, Wenatchee, WA, USA).

## 4. Conclusions

Two vibrational spectroscopic techniques—NIR and MIR spectroscopy—were applied for the discrimination of 93 Argan oil samples from four geographical origins (Agadir, Essaouira, Taroudant and Tiznit). Exploratory data analysis by PCA was carried out to visualize the geographical origin of the Argan oil samples by the PC plots. Further, two classification methods, PLS-DA and SIMCA, were tested. Standard preprocessing as well as multivariate filter techniques were applied to optimize both the PLS-DA and SIMCA classification models. Evaluating the applied preprocessing methods, based on the statistical parameters of specificity, sensitivity, precision and accuracy, it was seen that the multivariate filters provide better results (they can be ranked as GLSW = EPO > OSC). The three multivariate filtering methods positively influence the models, and, therefore, provide an accurate discrimination of the four classes of Argan oil.

The training dataset demonstrated similar classification results for the PLS-DA and SIMCA models, with 100% sensitivity, specificity, precision and accuracy for all classes. However, SIMCA models were unable to assign all test set samples to their predefined groups. Therefore, for both NIR and MIR datasets, the PLS-DA models outperform the SIMCA models in terms of prediction with 100% accuracy.

This study highlighted the benefit of using the MIR and NIR techniques as fast, non-destructive, non-invasive and low-cost analytical methods. These techniques, in combination with classification techniques and appropriate preprocessing methods, may replace the traditional techniques to classify Argan oil according to its geographical origin.

## Figures and Tables

**Figure 1 molecules-28-05698-f001:**
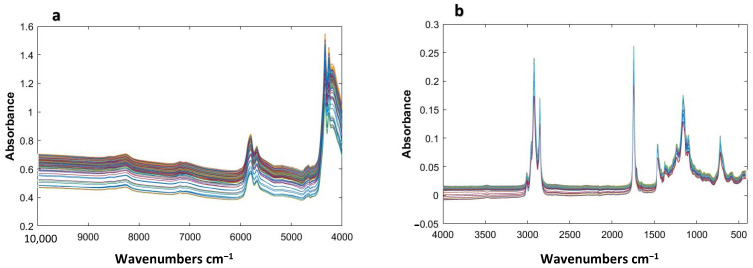
NIR (**a**) and MIR (**b**) raw spectra of the 93 Argan oil samples from four geographical origins.

**Figure 2 molecules-28-05698-f002:**
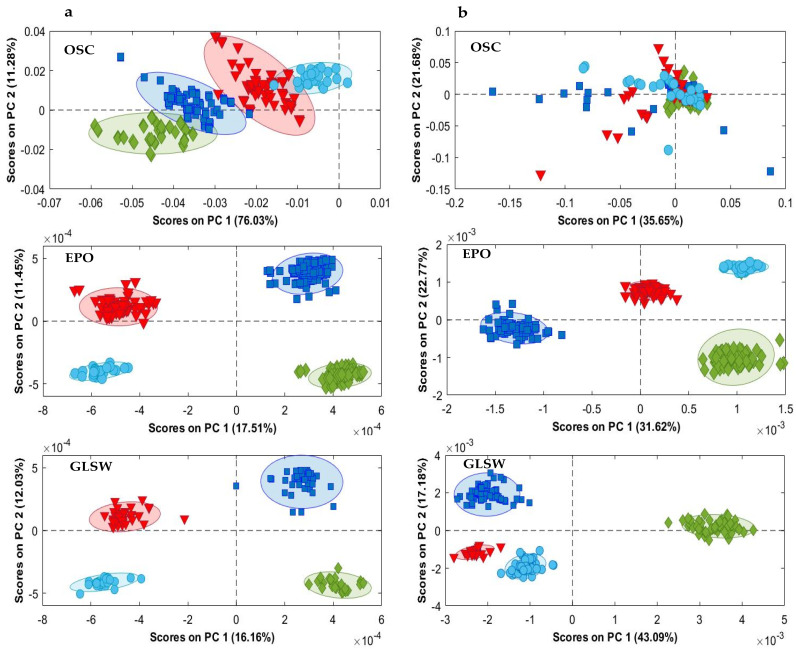
PCA score plots (PC1 vs. PC2) of the MIR (**a**) and NIR (**b**) data after applying a filter method. Taroudant (▼); Essaouira (♦); Agadir (■); Tiznit (●).

**Figure 3 molecules-28-05698-f003:**
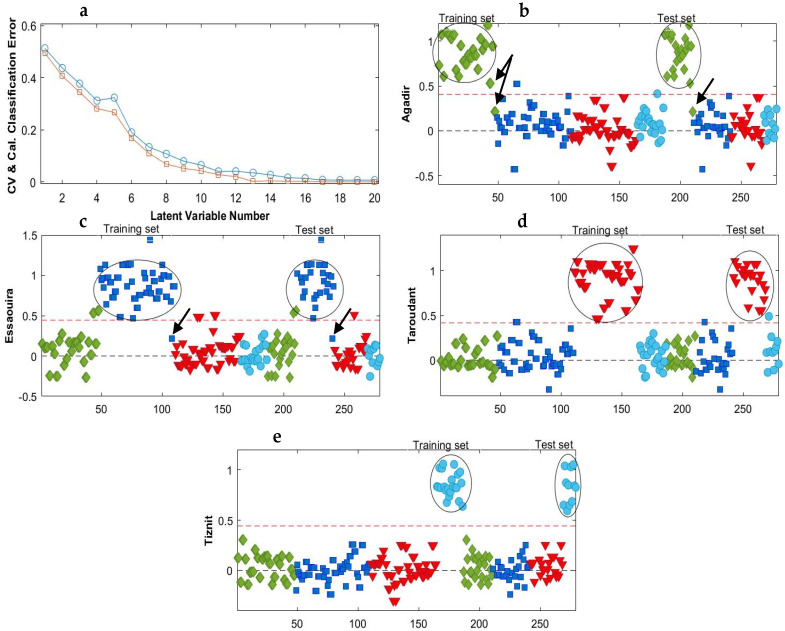
Average classification error as a function of model complexity (**a**) and the PLS-DA prediction results for the MIR data after smoothing + 2nd derivative and mean centering for (**b**) class Agadir (♦), (**c**) class Essaouira (■), (**d**) class Taroudant (▼) and (**e**) class Tiznit (●). The red dotted lines represent the classification threshold. In (**a**): CV (□): Cross-validation; Cal (○): Calibration.

**Figure 4 molecules-28-05698-f004:**
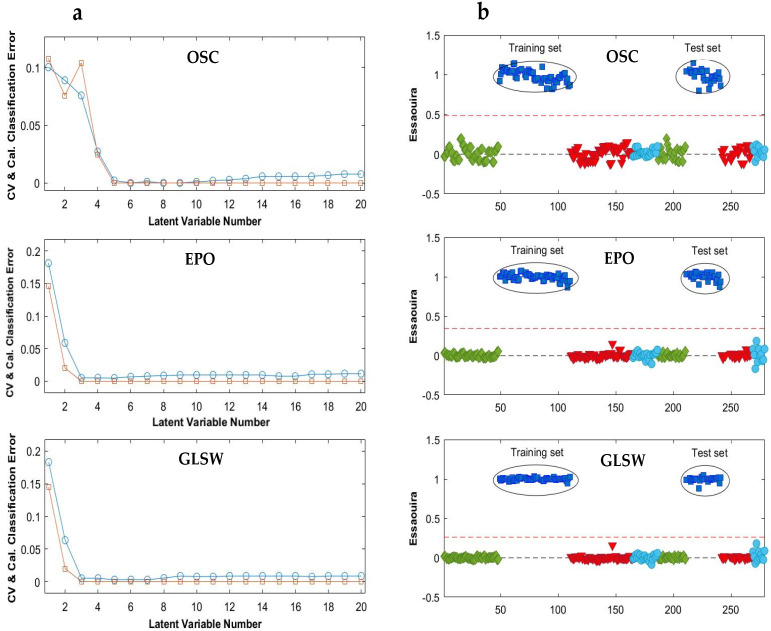
(**a**) Average classification errors as a function of model complexity after OSC, EPO and GLS-W preprocessing of the MIR data, and (**b**) the PLS-DA prediction results for the Essaouira group in the selected models (Table 2). Agadir (♦); Essaouira (■); Taroudant (▼); Tiznit (●); CV (○): Cross-validation; Cal (□): Calibration. The red dotted lines represent the classification threshold (or threshold value).

**Figure 5 molecules-28-05698-f005:**
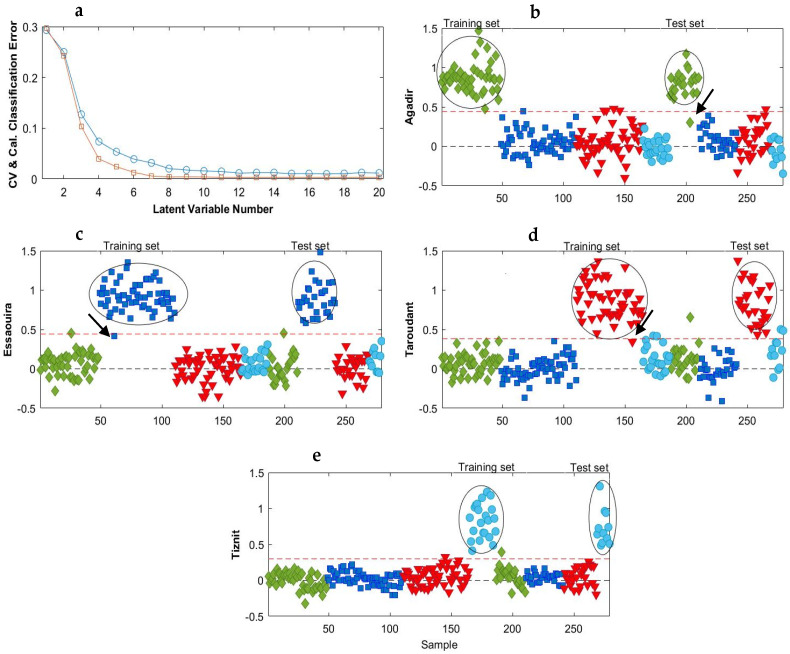
Average classification error as a function of model complexity (**a**) and the PLS-DA prediction results for the NIR data after MSC followed by 2nd derivative: for (**b**) class Agadir (♦), (**c**) class Essaouira (■), (**d**) class Taroudant (▼) and (**e**) class Tiznit (●). The red dotted lines represent the classification threshold. In (**a**), CV (○): Cross-validation; Cal (□): Calibration.

**Figure 6 molecules-28-05698-f006:**
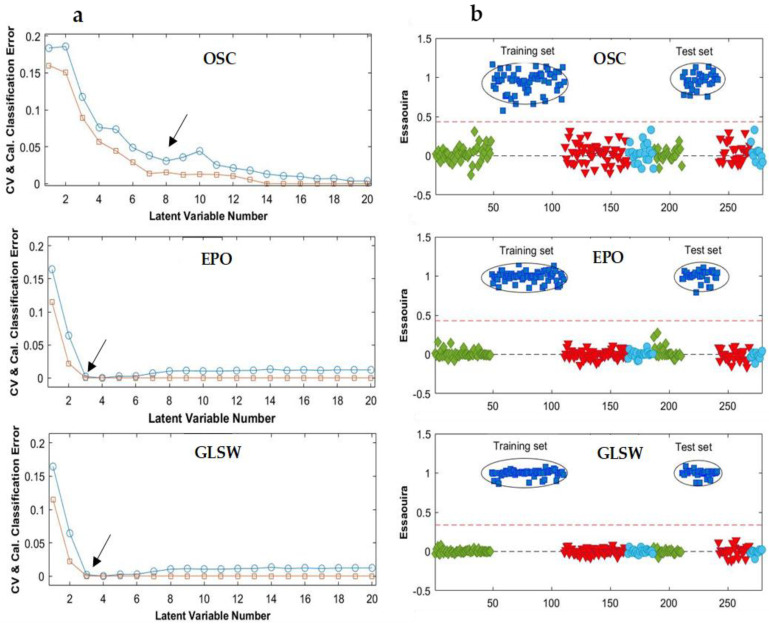
(**a**) Average classification error as a function of model complexity after OSC, EPO and GLS-W preprocessing for NIR data and (**b**) the PLS-DA prediction results of Essaouira group for the selected models (Table 3). Agadir (♦); Essaouira (■); Taroudant (▼); Tiznit (●). In (**a**), CV (○): Cross-validation; Cal (□): Calibration. The red dotted lines are related to the class threshold (or threshold value).

**Table 1 molecules-28-05698-t001:** NIR and MIR spectral regions of Argan oil and the corresponding functional groups.

FT-NIR		FT-MIR	
Wavenumber (cm^−1^)	Functional Group	Wavenumber (cm^−1^)	Functional Group
4350–4210	C–O, C–H	685	C–C
4800–4500	–HC=CH–	722	–HC=CH– (cis)
6000–5500	–CH_2_, –CH_3_	968	–HC=CH– (trans)
7180–7075	C–H	1097	–C–C
8300–8200	C–H	1118	–C–O
		1160	–CH_2_
		1238	–CH_2_
		1377	=C–H– (cis)
		1417	=C–H– (cis)
		1463	–C–H- (CH_2_, CH_3_)
		1744	–C=O
		2854	–C–H (CH_2_)
		2923	–C–H (CH_2_)
		3008	–C=C–H (cis)

**Table 2 molecules-28-05698-t002:** Results of Partial Least Squares-Discriminant Analysis on the MIR spectra of Argan oil samples, before and after different preprocessing. Spec: Specificity; Sens: Sensitivity; Prec: Precision; Acc: Accuracy.

Training Set			Test Set							
Preprocessing	Region	LVs	Sens (%)	Spec (%)	Prec (%)	Acc (%)	Sens (%)	Spec (%)	Prec (%)	Acc (%)
None	AG	7	56.6	75.8	49.1	70.2	69.5	80.6	57.1	77.6
	ES		51.7	86.9	65.2	75.7	62.9	89.6	73.9	81.2
	TA		48.8	95.5	78.6	82.3	50.0	96.7	85.7	83.5
	TZ		63.6	79.9	30.4	77.9	81.8	85.1	45.0	84.7
Autoscaling	AG	8	66.6	93.2	78.0	86.2	66.6	93.4	80.0	85.9
	ES		69.0	84.5	67.8	79.5	70.4	87.9	73.1	82.4
	TA		77.3	90.6	77.3	86.7	82.6	88.8	73.0	87.1
	TZ		100.0	96.2	78.6	96.7	90.9	95.9	76.9	95.3
Smoothing + 2nd derivative + Mean centering	AG	9	95.8	97.7	93.9	97.2	95.8	98.4	95.8	97.6
	ES		94.8	98.4	96.5	97.2	96.3	98.3	96.3	97.6
	TA		100.0	100.0	100.0	100.0	100.0	100.0	100.0	100.0
	TZ		100.0	100.0	100.0	100.0	100.0	100.0	100.0	100.0
OSC	AG	5	100.0	100.0	100.0	100.0	100.0	100.0	100.0	100.0
	ES		100.0	100.0	100.0	100.0	100.0	100.0	100.0	100.0
	TA		100.0	100.0	100.0	100.0	100.0	100.0	100.0	100.0
	TZ		100.0	100.0	100.0	100.0	100.0	100.0	100.0	100.0
EPO	AG	3	100.0	100.0	100.0	100.0	100.0	100.0	100.0	100.0
	ES		100.0	100.0	100.0	100.0	100.0	100.0	100.0	100.0
	TA		100.0	100.0	100.0	100.0	100.0	100.0	100.0	100.0
	TZ		100.0	100.0	100.0	100.0	100.0	100.0	100.0	100.0
GLSW	AG	3	100.0	100.0	100.0	100.0	100.0	100.0	100.0	100.0
	ES		100.0	100.0	100.0	100.0	100.0	100.0	100.0	100.0
	TA		100.0	100.0	100.0	100.0	100.0	100.0	100.0	100.0
	TZ		100.0	100.0	100.0	100.0	100.0	100.0	100.0	100.0

**Table 3 molecules-28-05698-t003:** Results of partial least squares-discriminant analysis on the NIR spectra of Argan oil samples, before and after different preprocessing. Spec: Specificity; Sens: Sensitivity; Prec: Precision; Acc: Accuracy.

			Training Set				Test Set			
Preprocessing	Region	LVs	Sens (%)	Spec (%)	Prec (%)	Acc (%)	Sens (%)	Spec (%)	Prec (%)	Acc (%)
None	AG	10	58.3	89.7	66.7	81.5	62.5	86.9	62.5	80.6
	ES		91.9	96.7	93.4	95.1	96.8	96.7	93.7	96.8
	TA		75.0	93.2	81.2	88.0	62.9	95.4	85.0	86.0
	TZ		72.7	89.5	48.5	87.5	54.5	86.5	35.3	82.8
Autoscaling	AG	8	64.6	88.4	65.9	82.8	70.8	84.1	60.7	80.6
	ES		96.8	99.1	98.3	98.4	100.0	96.8	93.9	97.8
	TA		77.8	91.7	79.2	87.5	66.6	92.4	78.3	84.9
	TZ		77.3	95.1	68.0	93.0	45.4	95.1	55.6	89.2
MSC + 2nd derivative	AG	6	100.0	99.4	95.6	99.4	95.8	100.0	100.0	98.9
	ES		99.3	100.0	96.8	99.6	100.0	100.0	100.0	100.0
	TA		98.1	100.0	100.0	99.4	100.0	100.0	100.0	100.0
	TZ		100.0	100.0	100.0	100.0	100.0	100.0	100.0	100.0
OSC	AG	8	100.0	100.0	100.0	100.0	100.0	100.0	100.0	100.0
	ES		100.0	100.0	100.0	100.0	100.0	100.0	100.0	100.0
	TA		100.0	100.0	100.0	100.0	100.0	100.0	100.0	100.0
	TZ		100.0	100.0	100.0	100.0	100.0	100.0	100.0	100.0
EPO	AG	3	100.0	100.0	100.0	100.0	100.0	100.0	100.0	100.0
	ES		100.0	100.0	100.0	100.0	100.0	100.0	100.0	100.0
	TA		100.0	100.0	100.0	100.0	100.0	100.0	100.0	100.0
	TZ		100.0	100.0	100.0	100.0	100.0	100.0	100.0	100.0
GLSW	AG	3	100.0	100.0	100.0	100.0	100.0	100.0	100.0	100.0
	ES		100.0	100.0	100.0	100.0	100.0	100.0	100.0	100.0
	TA		100.0	100.0	100.0	100.0	100.0	100.0	100.0	100.0
	TZ		100.0	100.0	100.0	100.0	100.0	100.0	100.0	100.0

**Table 4 molecules-28-05698-t004:** Results of SIMCA analysis on the MIR (**a**) and NIR (**b**) spectra of Argan oil after different preprocessing. Spec: Specificity; Sens: Sensitivity; Prec: Precision; Acc: Accuracy.

				Training Set				Test Set			
	Preprocessing	Region	PCs	Sens (%)	Spec (%)	Prec (%)	Acc (%)	Sens (%)	Spec (%)	Prec (%)	Acc (%)
(**a**) MIR	MSC + 1st derivative	AG	5	92	100	97	87	70	100	95	81
		ES	4	92	100	97	90	57	100	87	76
		TA	4	96	100	99	93	58	100	89	77
		TZ	3	77	100	100	94	45	100	89	84
	EPO	AG	4	100	100	100	100	89	100	100	91
		ES	5	100	100	100	100	96	100	100	94
		TA	4	100	100	100	100	83	100	100	94
		TZ	3	100	100	100	100	82	100	100	97
	GLSW	AG	4	100	100	100	100	85	100	100	92
		ES	6	100	100	100	100	77	100	100	95
		TA	5	100	100	100	100	87	100	100	95
		TZ	3	100	100	100	100	82	100	100	97
(**b**) NIR	MSC + 2nd derivative	AG	5	91	100	88	85	74	100	88	83
		ES	3	85	100	94	85	64	100	89	83
		TA	3	89	100	93	85	62	100	85	75
		TZ	8	77	100	97	86	54	100	89	85
	EPO	AG	4	100	100	100	100	89	100	100	97
		ES	4	100	100	100	100	81	100	100	97
		TA	4	100	100	100	100	92	100	100	95
		TZ	2	100	100	100	100	82	100	100	97
	GLSW	AG	3	100	100	100	100	89	100	100	98
		ES	3	100	100	100	100	84	100	100	96
		TA	4	100	100	100	100	92	100	100	95
		TZ	2	100	100	100	100	91	100	100	98

## Data Availability

Not applicable.

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
