# Peer review of "Evaluation of Multivariate Filters on Vibrational Spectroscopic Fingerprints for the PLS-DA and SIMCA Classification of Argan Oils from Four Moroccan Regions"

_molecules, 2023, doi:10.3390/molecules28155698_

Round 1

Reviewer 1 Report

The manuscript presents the results of a study aimed at developing a NIR and MIR spectroscopy-based method to classify Argan oil as a function of the geographical origin. In particular, several standard and multivariate pre-processing methods of the spectra were evaluated, and the performances of the developed models were assessed.

The manuscript is fairly well structured and presented. The scientific approach used by the authors is adequate, although the proposed approach of using NIR and MIR spectroscopy fingerprinting for authenticity purposes is widely documented in analytical chemistry. Therefore, the novelty of this study is the application of that approach to the discrimination of Argan oil based on geographical origin.

The major criticism is related to the number of samples (93 Argan oil samples, line 381), for each of them three independent (line 387). This means that the real different samples of this study were 93, and three replicates were considered. This choice has an impact on the way used to the authors to split the samples into training and test sets, the latter to validate the developed models. For this purpose, usually the whole dataset is split dividing the samples into two groups in ratio 80:20 (or 75:25), of which 80% (or 75%) of them are used as training set, and the remaining 20% (or 25%) are used as test set. This means that the samples used to validate the models were not included into the training set, therefore, not used to build the models. Can the authors motivate their choice in defining the training and test sets?

Some suggestions:

Check that the reference format is consistent with the requirements of Molecules Journal.

Line 512: software.

Reviewer 2 Report

In this paper, chemometric models were built to distinguish geographic origin of Morrocan argan oil using near and middle infrared spectroscopies. The article is adequate to be published in the Molecules bo but few recommendations were registered above.

Lines 134-136. Is associated the information about unclear distinction between four regions to raw or preprocessed data spectra?

Lines 175-176. “and hoped. These results, from both instruments, show that OSC, EPO and GLSW could be applied in practice and do not show groups due to chance correlations”. Can the authors apply the data fusion in the specific case in practice if the model not show groups?

The authors need to present the used GLSW, OSC and EPO preprocessing parameters. What window and polynomial order were used in the derivative Savitzky-Golay preprocessing?

Lines 140-141. What variable selection techniques were used to choice the spectral regions 3238-2696 cm−1 and 1863-400 cm−1? Or full spectra was used in the all built chemometric models?

Round 2

Reviewer 1 Report

The authors addressed all the comments and suggestions.